# Beneficial effects of cellular coinfection resolve inefficiency in influenza A virus transcription

**Jessica R. Shartouny**[1,2☉], **Chung-Young Lee**[1,2☉¤], **Gabrielle K. Delima**[1,2], **Anice C. Lowen**[1,2]*

**1** Department of Microbiology and Immunology, Emory University School of Medicine, Atlanta, Georgia, United States of America, **2** Emory Center of Excellence for Influenza Research and Response (Emory-CEIRR), Atlanta, Georgia, United States of America

☉ These authors contributed equally to this work.
¤ Current address: Department of Microbiology, School of Medicine, Kyungpook National University, Jung-gu, Daegu, Republic of Korea
* anice.lowen@emory.edu

**Data Availability Statement:** All relevant data are within the manuscript and its Supporting Information files.

**Funding:** The work was supported by funding from NIH/NIAID under the Centers of Excellence for

## Abstract

For diverse viruses, cellular infection with single vs. multiple virions can yield distinct biological outcomes. We previously found that influenza A/guinea fowl/Hong Kong/WF10/99 (H9N2) virus (GFHK99) displays a particularly high reliance on multiple infection in mammalian cells. Here, we sought to uncover the viral processes underlying this phenotype. We found that the need for multiple infection maps to amino acid 26K of the viral PA protein. PA 26K suppresses endonuclease activity and viral transcription, specifically within cells infected at low multiplicity. In the context of the higher functioning PA 26E, inhibition of PA using baloxavir acid augments reliance on multiple infection. Together, these data suggest a model in which sub-optimal activity of the GFHK99 endonuclease results in inefficient priming of viral transcription, an insufficiency which can be overcome with the introduction of additional viral ribonucleoprotein templates to the cell. More broadly, the finding that deficiency in a core viral function is ameliorated through multiple infection suggests that the fitness effects of many viral mutations are likely to be modulated by multiplicity of infection, such that the shape of fitness landscapes varies with viral densities.

## Author summary

Traditionally, a single virus particle is thought to be sufficient to initiate infection of a cell. However, the importance of coordinated infection, in which multiple viruses infect the same cell, has recently become apparent in diverse viral systems. With the goal of understanding how multiple infection increases the likelihood of productive infection, we investigated the traits that give rise to an extremely high reliance on multiple infection for influenza A/guinea fowl/HK/WF10/99 (H9N2) virus. We found that the phenotype maps to a single amino acid which results in a deficiency in transcription of viral mRNAs, a central function in the viral life cycle. This discovery suggests that the deleterious effects of

Influenza Research and Response contract no. 75N93021C00017 to ACL and R01 AI127799 to ACL. This research project was supported in part by the Emory University School of Medicine Flow Cytometry Core. The funders had no role in study design, data collection and analysis, decision to publish, or preparation of the manuscript.

**Competing interests:** The authors have declared that no competing interests exist.

many mutations that arise randomly within the influenza A virus genome may be alleviated when multiple viruses infect the same cell. In turn, coordinated infection is expected to be a major factor defining the evolutionary trajectories of viruses.

## Introduction

Cellular coinfection modulates the biology of diverse viruses. In VSV, inducing multiple infections via virion aggregation can accelerate viral production such that it outpaces innate antiviral responses [1]. In HIV-1, delivery of multiple viral genomes through cell-to-cell transmission of infection leads to more rapid onset of viral gene expression [2, 3]. For rotavirus and norovirus, vesicle-bound packets of viral particles have enhanced infectivity relative to free virions and are important vehicles for transmission [4, 5].

Similarly, in the case of influenza A viruses (IAVs), interactions between homologous coinfecting viruses can be highly biologically significant [6–8]. While most single infections are abortive, delivery of multiple viral genomes to a cell strongly increases the likelihood of productive infection and can augment both replication rate and yield [6, 7, 9]. In other words, the virus-virus interactions that play out during cellular coinfection are typically beneficial and often required for productive infection. Of note, this feature of IAV biology strongly increases the frequency of reassortment, an important source of viral genetic diversity [10].

IAV reliance on multiple infection appears to be particularly acute under conditions that are unfavorable for viral replication, such as in a new host species [7]. For a G1-lineage strain, influenza A/guinea fowl/Hong Kong/WF10/99 (H9N2) virus (GFHK99, also referred to as WF10), we found that coinfection with multiple homologous viruses was essential for robust replication in mammalian cells, but not avian cells. While every IAV tested thus far has shown some reliance on multiple infection, the strong host-specific reliance of GFHK99 was not apparent for influenza A/mallard/Minnesota/199106/99 (H3N8) virus (MaMN99). Using gene segment reassortments between these two IAVs, we found that the PA segment drives the high reliance of GFHK99 virus on multiple infection. However, the functional basis for this genetic association remained unclear.

The PA gene segment encodes two proteins: PA and PA-X. PA is one of three protein subunits that comprise the viral RNA-dependent RNA polymerase, along with PB2 and PB1. The N-terminus of PA contains an endonuclease that cleaves cellular mRNAs 10–20 bases downstream of the 5' cap, in a process termed cap-snatching [11, 12]. The resultant capped primer is required for mRNA synthesis by the viral polymerase [13]. PA-X, discovered in the past decade, is produced by ribosomal frame-shifting during translation of the PA mRNA [14]. The N-terminal 191 amino acids of PA-X are identical to those of PA, while the C-terminal 41 or 61 amino acids are unique [15]. PA-X has been shown to contribute to the shutoff of host protein synthesis [16].

In this work, we sought to elucidate the drivers of the GFHK99 strain's high reliance on multiple infection, tied to the PA gene segment. Our data demonstrate that the phenotype maps to residue 26K within the PA endonuclease. This amino acid lowers endonuclease activity, leading to inefficient viral transcription in cells infected at low multiplicity of infection (MOI). Conditions conducive to cellular coinfection allow robust transcription by a PA 26K virus in mammalian cells and support enhanced progeny production. Treatment of infected cells with a PA endonuclease inhibitor, baloxovir acid, leads to a similar reliance on multiple infection. These data suggest that cellular coinfection is beneficial because the delivery of multiple copies of the eight viral ribonucleoproteins (vRNPs) to the cell increases the frequency of

successful transcription events, allowing infection to be initiated efficiently despite sub-optimal conditions.

## Results

### Reliance on multiple infection of GFHK99 is linked to the endonuclease region of PA

To determine which region of the PA gene segment contributed to the high reliance on multiple infection seen in GFHK99, chimeric PA gene segments were created using GFHK99 PA and MaMN99 PA. The segment was divided into three regions roughly corresponding to the domains of the PA protein (Fig 1A). Each chimeric segment and the full-length GFHK99 PA were incorporated into the MaMN99 background. To allow virus-virus interactions to be monitored, homologous infection pairs, termed wildtype (WT) and variant (VAR) viruses, were produced for each PA genotype. VAR viruses contained a synonymous mutation in each of the eight gene segments to allow differentiation from WT segments in molecular assays. In addition, WT and VAR virus HA proteins were differentially modified with epitope tags to allow quantification of cellular infection at a protein level [7].

Using flow cytometry, the frequency of cellular coinfection between WT and VAR viruses was evaluated. Since this assay detects viral protein, it measures the extent to which viral gene expression relies on multiple infection. MDCK cells were coinfected with homologous WT and VAR pairs and, to enable quantitative analysis, infections were limited to a single cycle such that progeny viruses cannot be propagated onward. The relationship between the percentage of cells that stained with one or both epitope tags (total HA$^+$) and the percentage of cells staining with both tags (dual-HA$^+$) was evaluated (Fig 1B). As seen previously, MaMN99 WT and VAR viruses produced distinct populations of singly-infected cells and coinfected cells, with the frequency of coinfection increasing with total infection levels. Conversely, for the GFHK99 PA in a MaMN99 background (GFHK99 PA virus), nearly all infected cells were coinfected with both WT and VAR. Thus, a linear relationship was seen between dual-HA$^+$ and total HA$^+$, with a slope of 1.02 (95% C.I. 0.979 to 1.07, $R^2$ = 0.981). Both the GFHK99 Arch PA and GFHK99 C-term PA viruses showed similar infection patterns to that of MaMN99 virus. In contrast, the GFHK99 Endo PA virus displayed a linear relationship between dual-HA$^+$ and total HA$^+$, like the GFHK99 PA virus. The slope obtained from a linear regression of GFHK99 PA Endo was comparable to that of GFHK99 PA at 1.03 (95% C.I. 0.959 to 1.10, $R^2$ = 0.971).

The prevalence of reassortant viruses within the progeny virus population was then determined. Since only cells coinfected with WT and VAR viruses can produce reassortants, this assay gives an indication of the relative productivity of singly- and multiply-infected cells. Reassortants were identified by deriving clonal isolates from the progeny population and then genotyping each segment therein. The frequency of reassortants within MaMN99 progeny virus populations was low at lower infection levels and increased as the percentage of HA$^+$ cells increased. By comparison, GFHK99 PA virus coinfections resulted in high frequencies of reassortment even at low levels of infection, signifying that most of the progeny viruses were produced in cells infected with both WT and VAR (Fig 1C). WT-VAR coinfections with GFHK99 Arch and GFHK99 C-term viruses exhibited similar reassortment outcomes as seen with MaMN99 virus. Conversely, GFHK99 Endo PA coinfections displayed high percentages of reassortant progeny, on par with those seen for GFHK99 PA. The high frequencies of reassortant progeny and high levels of dual HA positivity observed in coinfections with GFHK99 Endo PA viruses indicate that the endonuclease region of GFHK99 PA confers a high reliance on multiple infection.

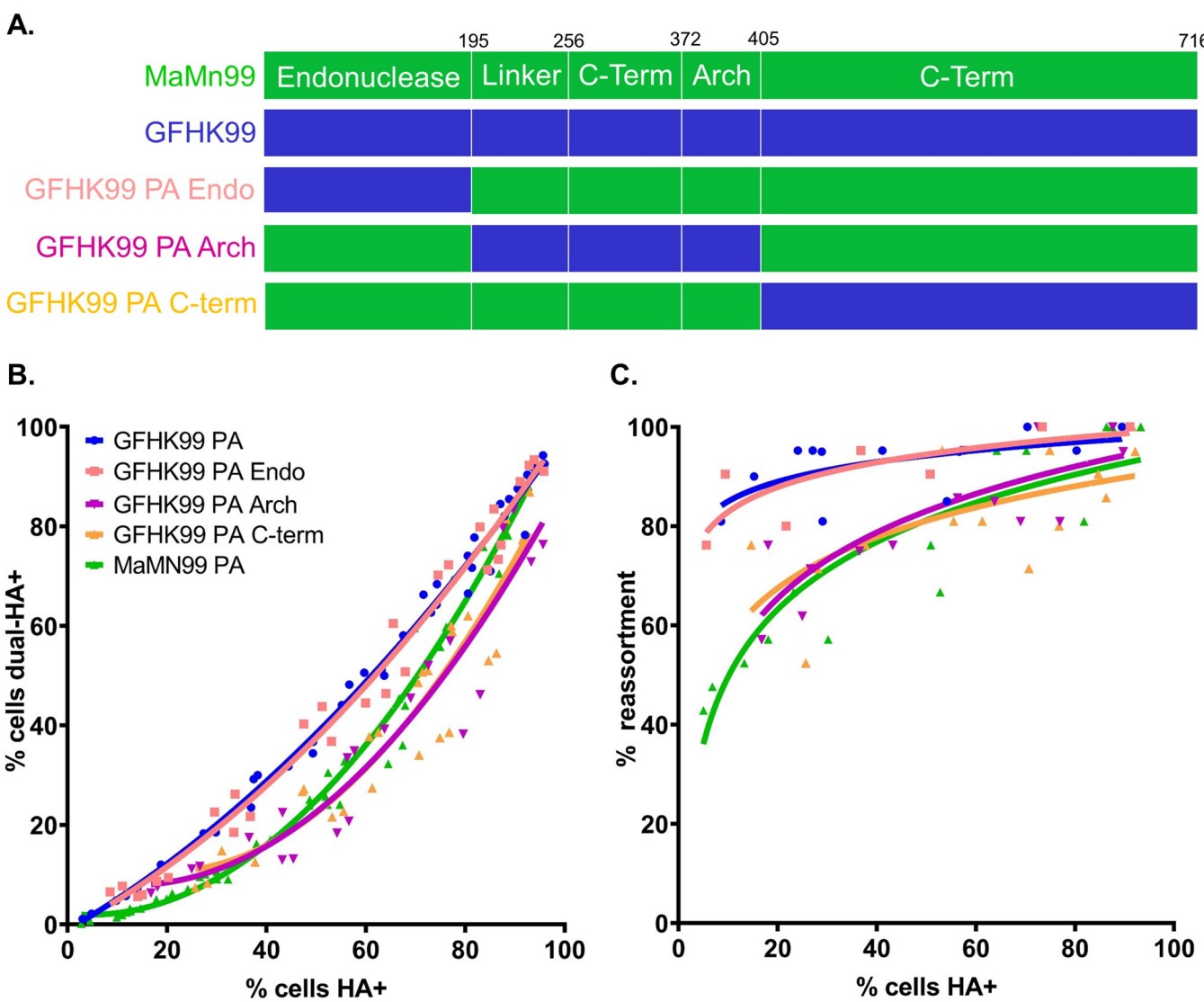

**Fig 1. Replacement of the MaMN99 PA endonuclease region with that of GFHK99 PA increases reliance on multiple infection.** (A) Chimeric PA gene segments that introduce regions of the GFHK99 PA into the MaMN99 PA: the endonuclease (GFHK99 Endo), the middle region (GFHK99 Arch), and the C-terminal domain (GFHK99 C-term). Amino acid positions are noted at the top of the figure. (B) The relationship between cells dually-infected with WT and VAR and cells infected with WT, VAR, or both. Data shown for each virus are derived from three independent experiments. Curve similarity to GFHK99 PA is determined by least sum-of-squares F test: GFHK99 PA Endo is represented by the same curve (p = 0.9385, F = 0.1356) while MaMN99, GFHK99 PA Arch and GFHK99 C-term are represented by different curves (p < 0.0001, F = 47.59, 34.24, and 39.01) (C) The percentage of progeny viruses with any reassortant genotype is plotted against the percentage of cells expressing hemagglutinin (HA). Curve similarity to GFHK99 PA is determined by least sum-of-squares F test: GFHK99 PA Endo is represented by the same curve (p = 0.8962, F = 0.11.0) while MaMN99, GFHK99 PA Arch and GFHK99 C-term are represented by different curves (p < 0.001, F = 23.73, 9.789, and 18.11) Data shown for each virus are derived from two independent experiments. The PA segment genotype of WT and VAR viruses is indicated in the legend. All viruses carried the remaining seven segments from MaMN99 virus.

## Disruption of PA-X does not alter reliance on multiple infection

The endonuclease domain is shared by the PA and PA-X proteins. To determine whether PA-X was the driving force behind high reliance on multiple infection, viruses were created in which the PA-X reading frame was disrupted [16]. Because PA-X is difficult to detect by western blotting, the effectiveness of this disruption was verified at a functional level. Using plasmid transfection, the full length MaMN99 PA gene segment with or without mutations to PA-X was introduced into cells together with a *Renilla* luciferase reporter construct. The wild

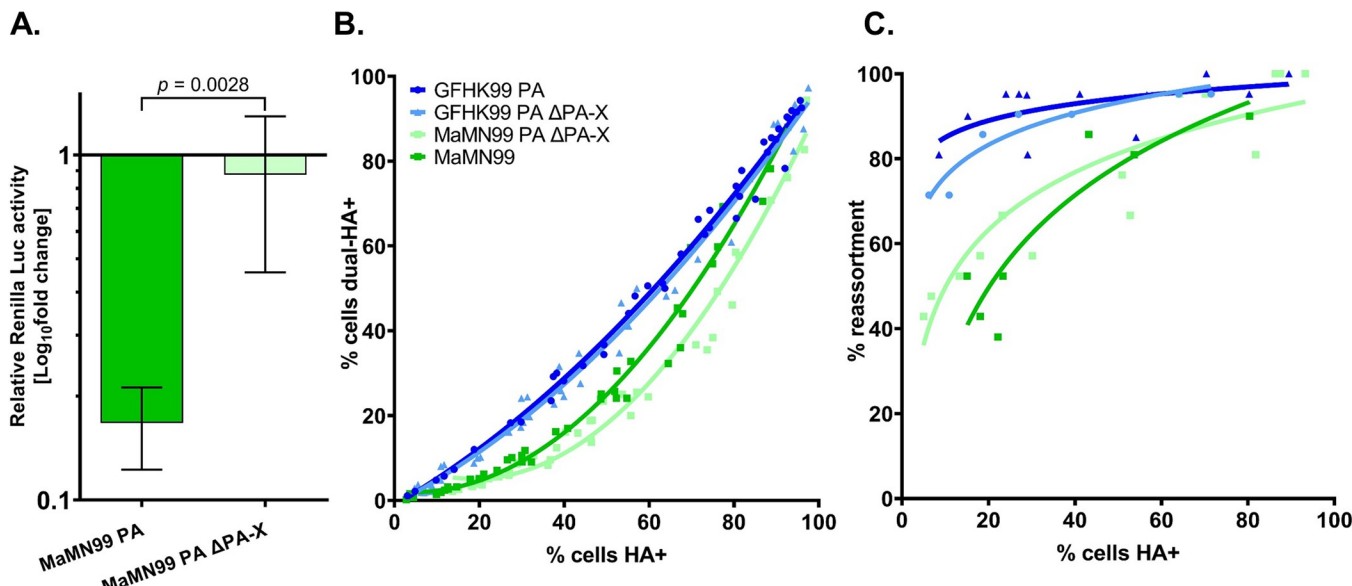

**Fig 2. Disrupting PA-X does not alter reliance on multiple infection of GFHK99 and MaMN99 viruses.** (A) Relative luciferase activity of cells co-transfected with a *Renilla* luciferase expression plasmid and either MaMN99 PA or MaMN99 PA ΔPA-X expression plasmids. Values represent means ± SD of three replicates. Statistical significance was analyzed by unpaired t-test. (B) The relationship between cells dually-infected with WT and VAR MaMN99 ΔPA-X or GFHK99 PA ΔPA-X (dual-HA+) and cells infected with WT, VAR, or both (HA+). Data plotted are from three independent experiments. Curve similarity to GFHK99 PA is determined by least sum-of-squares F test: GFHK99 PA ΔPA-X is represented by a similar curve (p = 0.38, F = 1.05) while MaMN99 ΔPA-X is represented by a different curve (p < 0.0001, F = 51.4) (C) The percentage of progeny viruses with any reassortant genotype is plotted against the percentage of cells expressing hemagglutinin (HA). Curve similarity was determined by least sum-of-squares F test: GFHK99 PA ΔPA-X is represented by a similar curve to GFHK99 PA (p = 0.11, F = 2.8) while MaMN99 ΔPA-X is represented a similar curve to MaMN99 (p = 0.059, F = 3.4). Data shown are derived from two independent experiments. The PA segment genotype of WT and VAR viruses is indicated in the legend. All viruses carried the remaining seven segments from MaMN99 virus. Results for GFHK99 PA and MaMN99 viruses are reproduced from Fig 1 for comparison.

type PA construct strongly suppressed the reporter signal relative to that seen with the ΔPA-X construct, consistent with the host shut-off activity of PA-X [16] and confirming the effectiveness of the mutations introduced (Fig 2A).

WT and VAR homologous pairs of these viruses were used to coinfect MDCK cells and the frequencies of coinfected cells and reassortant viruses were examined across a range of MOIs (Fig 2). By both measures, MaMN99 ΔPA-X and GFHK99 PA ΔPA-X viruses displayed similar coinfection reliance phenotypes to their respective parental strain. Since expression of PA-X did not modulate the extent of reliance on multiple infection in either strain, PA-X does not appear to be a major driver of the high reliance phenotype displayed by GFHK99.

### Reliance on multiple infection is dependent on PA 26 in the endonuclease region

An alignment of the PA endonuclease regions of MaMN99 and GFHK99 showed five coding differences at PA amino acids 20, 26, 85, 101, and 118 (Fig 3A). Owing to the charge difference at PA 26, where MaMN99 has a glutamic acid (E) and GFHK99 has a lysine (K), we focused on this position, which is located proximal to the active site of the endonuclease region (Fig 3B). Reciprocal mutants were generated in the MaMN99 and GFHK99 PA segments and each was incorporated into the MaMN99 background. WT and VAR homologous pairs of MaMN99 PA E26K and MaMN99: GFHK99 PA K26E (GFHK99 PA K26E) viruses were used in coinfections.

The results showed that swapping the amino acid at PA 26 swapped the phenotypes displayed by MaMN99 and GFHK99 PA. Introduction of K26E within the GFHK99 PA

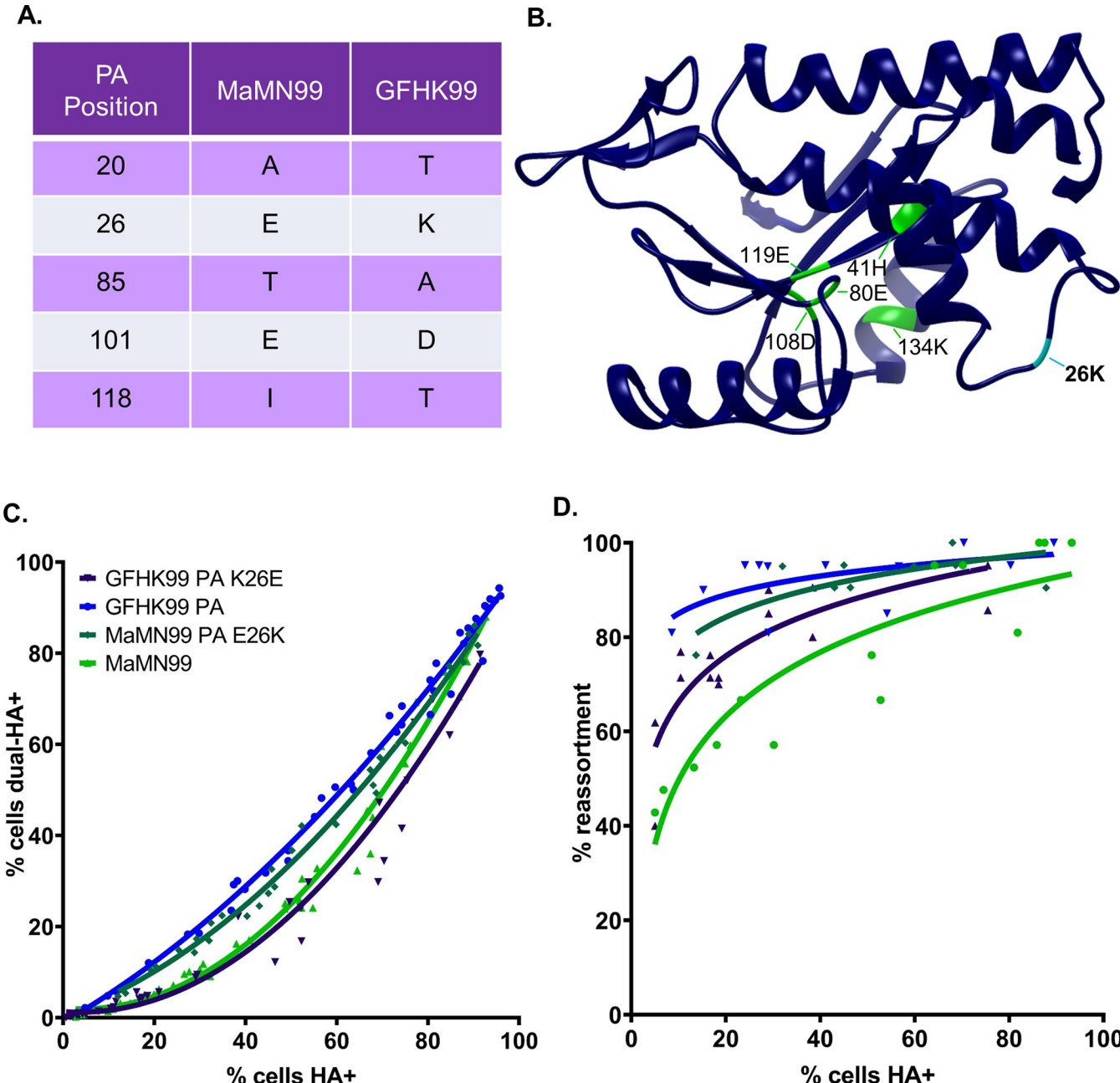

**Fig 3. PA 26K is associated with a higher reliance on multiple infection than PA 26E.** (A) The MaMN99 and GFHK99 PA endonuclease regions differ by five amino acids. (B) A model of the endonuclease domain of GFHK99 PA with the active site residues labeled in green and PA 26K labeled in cyan. (C) The relationship between cells dually-infected with WT and VAR (dual HA+) and cells infected with WT, VAR, or both (HA+). Data plotted are from three independent experiments. Significance of differences between the curves was evaluated by a least sum-of-squares F test: GFHK99 PA E26K differs from GFHK99 PA (p < 0.0001, F = 65.5) and MaMN99 E26K differs from MaMN99 (p< 0.0001, F = 18.8). (C) The percentage of progeny viruses with any reassortant genotype is plotted against the percentage of cells expressing hemagglutinin (HA+). Similarity between curves was assessed by least sum-of-squares F test: GFHK99 PA E26K differs from GFHK99 PA (p = 0.0017, F = 8.7) and MaMN99 E26K can be represented by a similar curve as GFHK99 PA (p = 0.078, F = 9.16). Data plotted are from two independent experiments. The PA segment genotype of WT and VAR viruses is indicated in the legend. All viruses carried the remaining seven segments from MaMN99 virus.

**Table 1. High reliance on multiple infection is associated with increased ratio of genome copies to infectious units.**

| Virus | Mean genome copy/PFU ratio (+/- SD)[1] |
|---|---|
| MaMN99 | 22.6 (0.51) |
| GFHK99 PA | 97.9 (51) |
| GFHK99 PA K26E | 26.3 (9.8) |

[1]Genome copy / PFU ratio was determined for three replicate samples of virus stock

decreased frequencies of dually HA+ cells, while introduction of E26K into the MaMN99 PA had the opposite effect (Fig 3C). In line with the coinfection results, GFHK99 PA K26E virus yielded fewer reassortants than GFHK99 PA virus, while MaMN99 PA E26K infection progeny were predominantly reassortant (Fig 3D). Thus, in both PA backgrounds, PA 26K was associated with higher reassortment than PA 26E. Taken together, the data show that viral gene expression and progeny production were focused within coinfected cells to a greater extent for viruses encoding PA 26K compared to those encoding PA 26E.

## High reliance on multiple infection is associated with increased ratio of genome copies to infectious units

A classical means of evaluating the efficiency of viral infection is to determine the specific infectivity; that is, the relationship between infectious units and a measure of physical viral particles such as protein or RNA content. To relate the viral phenotypes observed during coinfection to specific infectivity, we determined the ratio of genome copy number to plaque forming units (PFU) for MaMN99, GFHK99 PA, and GFHK99 PA K26E viruses (Table 1). As anticipated, each infectious unit of GFHK99 PA virus was associated with a higher number of genome copies than the other two strains.

## Endonuclease activity and transcript production are suppressed by PA 26K

We reasoned that the high reliance on multiple infection resulting from PA 26K might be a result of reduced PA protein levels in infected cells or impeded functionality of the PA protein. To evaluate the first possibility, PA protein levels in cells infected with GFHK99 PA or GFHK99 PA K26E viruses were compared by western blotting (Fig 4A). GFHK99 PA virus did not display less PA protein than GFHK99 PA K26E virus, indicating that PA 26K did not reduce the accumulation of PA during infection.

To evaluate viral endonuclease activity of MaMN99 and GFHK99 PA, we assayed the extent to which the corresponding PA-X proteins disrupted expression of *Renilla* luciferase in transfected cells. This approach was used because PA and PA-X carry the same endonuclease domain, but PA-X activity is more readily monitored in a cell-based assay. As expected based on the mRNA-degrading function of PA-X, the activity of *Renilla* luciferase gradually decreased with an increase in the amount of PA-X plasmid introduced into cells (Fig 5A). Of note, the effect was markedly weaker with the GFHK99 PA-X than the MaMN99 PA-X, indicating that the GFHK99 PA-X has lower endonuclease activity. Next, E26K or K26E variants of MaMN99 and GFHK99 PA-X, respectively, were tested and PA-X expression of each was verified by Western blot. The E26K mutation in MaMN99 PA-X reduced PA-X activity, showing less reduction of *Renilla* luciferase activity than the wild-type (Fig 5B and 5D). Conversely, the K26E mutation in GFHK99 PA-X enhanced PA-X activity (Fig 5C and 5E). Thus, position 26 within the viral endonuclease modulates its enzymatic activity.

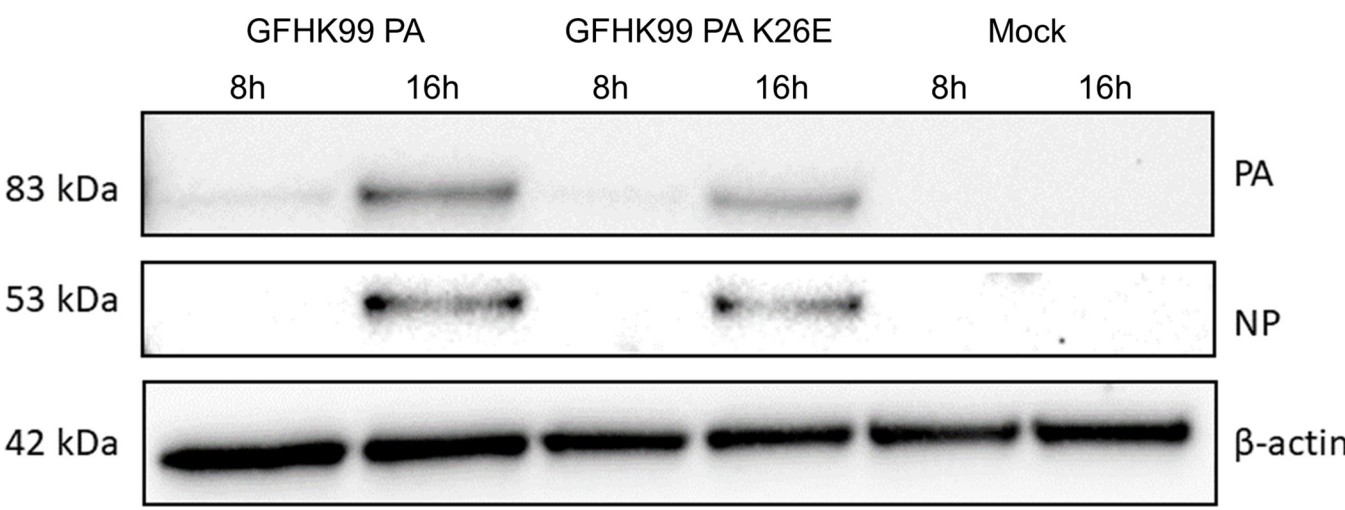

**Fig 4. Levels of PA protein are not decreased by PA 26K.** Western blot of cell lysates collected at 8 or 16 h post-infection with GFHK99 PA virus, GFHK99 PA K26E virus, or mock infected, as indicated above the blot. Blot was probed for PA, NP, and β-actin.

We further measured the effect of the E26K mutation in MaMN99 PA on viral transcription during infection by measuring the accumulation of viral mRNA under low and high MOI conditions in MDCK cells. At a low MOI, markedly less mRNA was produced in MaMN99--PA-E26K infection compared to MaMN99 infection (Fig 5F). However, at a high MOI, mRNA accumulated at a similar rate for MaMN99 and MaMN99-PA-E26K viruses (Fig 5H). These data suggest that low endonuclease activity of PA carrying the 26K polymorphism suppresses viral transcription at a low MOI but is compensated by multiple infection. Measurement of viral genomic RNA (vRNA) in the same cells (Fig 5G and 5I) revealed similar patterns, indicating that the effects of PA 26K and multiple infection are also borne out at the level of viral genome replication, most likely as a downstream consequence of the effects on viral transcription.

## Inhibition of endonuclease activity increases reliance on multiple infection

We postulated that inhibiting the PA endonuclease cap-snatching function would enforce an increased reliance on cellular coinfection for productive replication. The drug baloxavir marboxil (Xofluza) targets cap-snatching by chelating the ions in the active site of the PA protein and baloxavir acid (BXA), the active form of the drug, is available for use in cell culture. MaMN99 WT and VAR virus coinfections were treated with an intermediate dose of BXA designed to handicap but not wholly abolish infection (Fig 6A). The frequency of reassortant progeny resulting from these infections increased across the range of MOIs tested, indicating that nearly all progeny arose from coinfected cells under BXA treatment. This outcome is in stark contrast to that seen in mock-treated MaMN99 infections, where frequencies of reassortment suggest that appreciable levels of virus emanate from singly infected cells.

## Discussion

Although cellular coinfection can strongly impact infection outcomes in many virus-host systems, the mechanistic basis for these effects is generally poorly understood. Here we sought to address this deficiency by focusing on GFHK99, an IAV that shows extremely high reliance on cellular coinfection. We identified a polymorphism at position 26 within the PA endonuclease domain as the driver of high reliance on multiple infection. Functionally, PA 26K reduces the

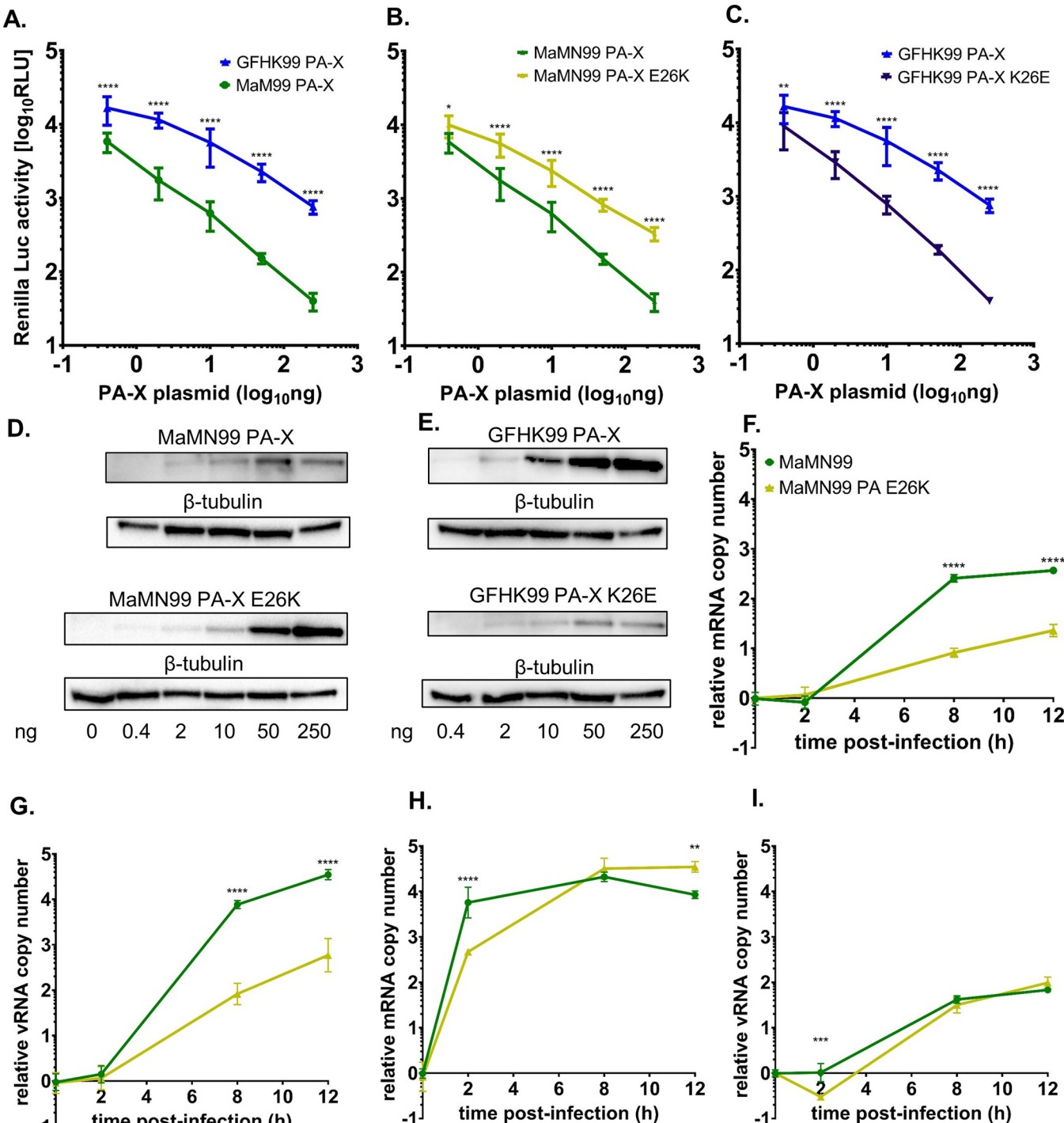

**Fig 5. 26K confers lower endonuclease activity than PA 26E. PA** (A-C) Dose-dependent effect of PA-X protein on expression of Renilla luciferase in co-transfected cells. Plots include data from three independent experiments, compared via 2-way ANOVA. ****$p < 0.0001$, *$p = 0.020$, **$p = 0.0017$. (A) MaMN99 PA-X and GFHK99 PA-X. (B) GFHK99 PA-X and MaMN99 PA-X E26K. (C) MaMN99 PA-X and MaMN99 PA-X E26K. (D-E) Representative western blots confirming expression of PA-X in transfected cells. Samples correspond to one of the replicates shown in panels A–C. Amount of PA-X plasmid transfected is indicated underneath (ng). (F-I) Quantification of viral mRNA (F, H) and vRNA (G, I) in cells infected at low MOI of 0.5 RNA copies/cell (F,G) and high MOI of 1000 RNA copies/cell (H, I). The mean $\log_{10}$ fold change relative to the 0 h time point is plotted, with error bars showing SD. vRNA and mRNA levels were compared via unpaired Student's t-test. ****$p < 0.0001$, ***$p = < 0.001$, **$p = < 0.01$.

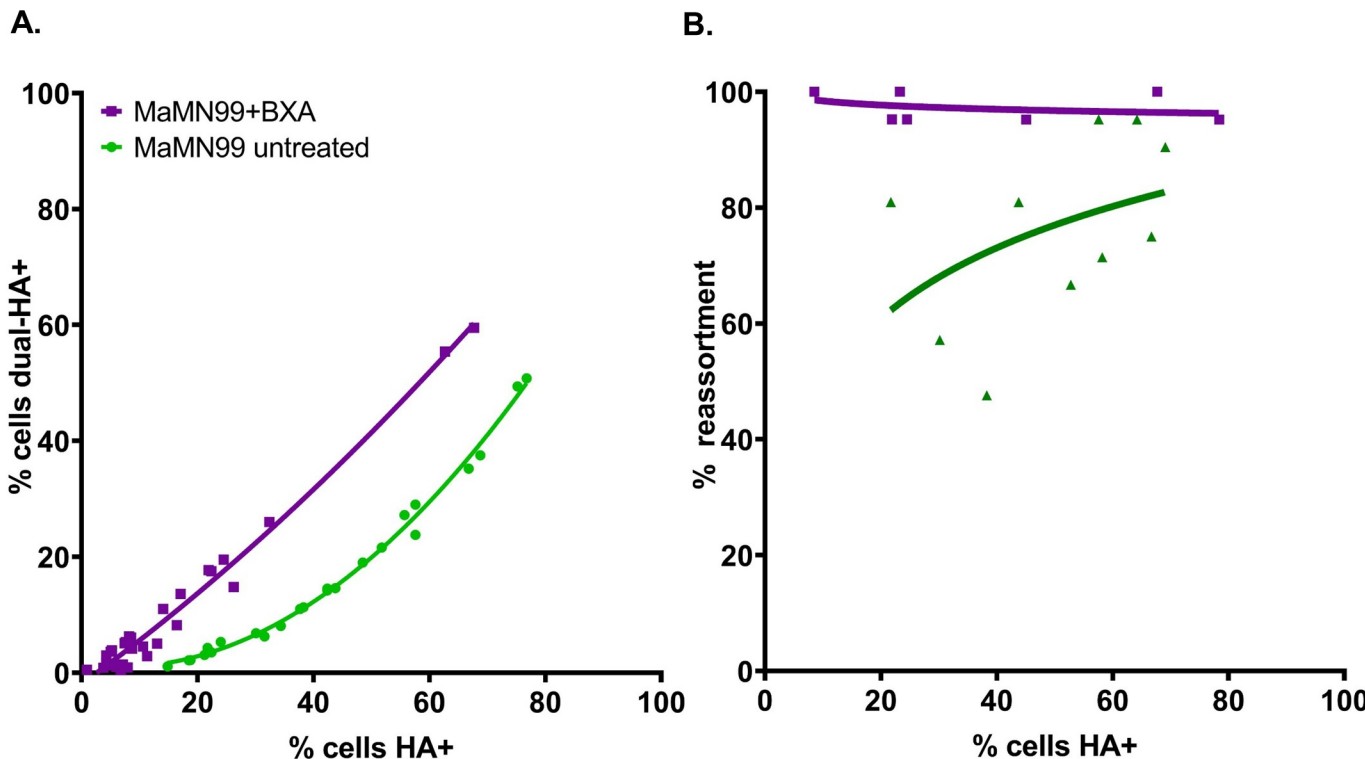

**Fig 6. Inhibition of cap-snatching increases reliance on multiple infection.** The MaMN99 PA endonuclease was inhibited using baloxavir acid (BXA) during a coinfection with homologous WT and VAR viruses. (A) The relationship between percentages of HA+ cells and dual-HA+ cells. The curve representing BXA-treated MaMN99 differs from that representing the untreated virus (p< 0.0001, F = 167.0, least sum-of-squares F test) (B) The percentage of progeny with reassortant genotypes was plotted as a function of percent cells HA+. The curve representing BXA-treated MaMN99 differs from that representing the untreated virus (p = 0.0077, F = 7.54, least sum-of-squares F test). Data from two independent experiments are plotted together.

activity of the endonuclease, lowering the efficiency of viral transcription and impacting the host shutoff capabilities of the virus. Importantly, however, under high MOI conditions, the inefficiency is overcome. Consistent with these results, BXA-imposed inhibition of the PA endonuclease leads to a focusing of viral replication within cells that are multiply-infected. Our data suggest a model in which the delivery of many viral genomes to the cell compensates for inefficient cap-snatching by providing greater opportunity for that process to unfold (Fig 7).

To successfully propagate its genome, an infecting virus must complete a number of discrete steps of the early life cycle, many of which are the targets of cellular defense mechanisms. For IAV, each barrier must be overcome by eight unique vRNP complexes. This process appears to be prone to failure, such that the likelihood of a single IAV establishing an infection is extremely low [9, 17]. Consistent with low rates of productive infection, IAV infected cells typically support expression from and replication of fewer than eight segments [6, 9]. In the case of GFHK99 virus, incomplete viral genomes were detected with a frequency of approximately 90% [7]. While high, this frequency is not sufficient to account for the extremely high reliance on multiple infection exhibited by this strain in mammalian systems [7]. The data reported here indicate that this additional reliance stems from a deficiency in viral cap-snatching, the process by which the IAV polymerase acquires capped RNA oligomers to prime transcription [12]. With multiple viral genomes infecting together, the larger number of NP-RNA templates and associated polymerases available for primary transcription appears to enable levels of mRNA synthesis sufficient to sustain robust infection.

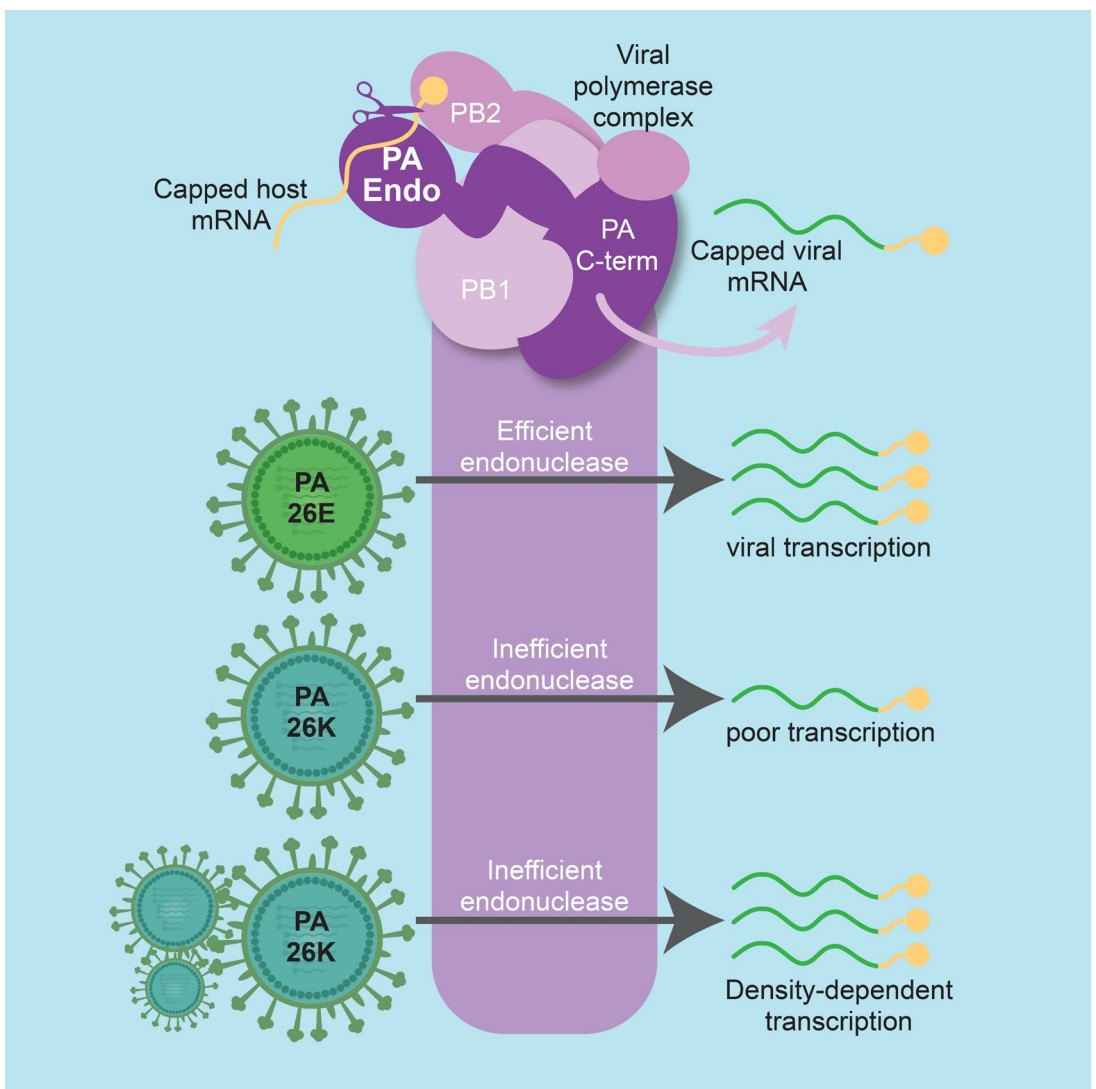

**Fig 7. Multiple infection allows a virus with an inefficient PA endonuclease to replicate.** The PA protein, located within the viral polymerase complex, cleaves host mRNAs to generate the capped primers needed for viral transcription. Our data show that PA 26K has low cleavage activity compared with PA 26E. Under low multiplicity conditions, this deficiency results in inefficient viral transcription. However, delivery of multiple copies of the viral genome to the cell allows a PA 26K polymerase to produce high levels of viral mRNA, enabling successful completion of the viral life cycle. Figure was created with BioRender. com.

Our previous work revealed that the extent to which GFHK99 virus relies on multiple infection for productive infection is strongly dependent on host species: in avian systems, the need for multiple infection was greatly diminished [7]. Since we focused our studies herein on mammalian systems, our data do not reveal whether the GFHK99 PA enzyme is more active in avian cells or whether this avian virus is simply less sensitive to the cap-snatching deficiency in a host environment to which it is well-adapted. However, prior work suggests the latter model is correct and that PA 26K is deleterious in both mammalian and avian hosts. First, PA 26K does not fall within interacting surfaces of defined cellular binding partners that might differ with host species. Second, PA 26K occurs rarely in sequenced strains, including those of

the G1 lineage of H9N2 viruses [18], suggesting its occurrence in GFHK99 is the result of genetic drift. Third, introduction of PA K26E was seen to improve the kinetics and yield of GFHK99 viral replication in chicken cells and eggs [18]. Fourth, GFHK99 variant viruses carrying PA K26E rapidly emerged and underwent positive selection in experimentally infected quail [19]. Thus, while GFHK99 shows a reduced need for multiple infection in avian cells compared to mammalian cells [7], there is a fitness cost of PA 26K even in avian systems. We therefore suggest that PA 26K has a greater impact in mammalian systems owing to a confluence of mal-adaptions to the mammalian cellular environment.

In using BXA treatment to substantiate the link between deficient cap-snatching and the need for many incoming viral genomes, we show that viral replication can proceed under antiviral treatment within multiply-infected cells. While potentially consequential for clinical use of baloxavir marboxil and the development of resistance [20, 21], our data more broadly suggest that the targeting of viral cap-snatching and other steps of the viral life cycle that precede genome replication are likely to be efficient therapeutic strategies. This approach is expected to increase the frequency of abortive infections, potentially pushing the rate of productive infections below a threshold needed to sustain viral propagation.

The viral population dynamics documented herein are consistent with the ecological principle of positive density dependence and, more specifically, the Allee effect. Positive density dependence is a situation in which increasing population density leads to a higher population growth rate [22]. An Allee effect refers specifically to positive density dependence within low density populations (such as in a low MOI viral infection). Allee effects occur when the members of a species rely on intra-species interactions for survival or reproduction [23, 24]. For example, predators may be more effective in packs and pollination may be more efficient when individual plants are in close proximity [25, 26]. A similar dynamic appears to apply within low density IAV populations: as more viruses are added to a cell, burst size increases. We propose that the positive relationship between initial viral input and ultimate viral output arises because essential steps of the early viral life cycle are less prone to stochastic failure when the infecting population is larger. In unfavorable contexts—such as a new host species, antiviral treatment or deleterious mutation—the frequency with which these essential steps fail may be increased. In these situations, viral density would be expected to have a greater impact on viral growth rates. The acute density dependence of GFHK99 in mammalian systems is well explained by this paradigm and we expect these concepts apply broadly to IAV and many other viruses.

This work reveals that, in the specific case of GFHK99, multiple infection mitigates the deleterious effects of a specific mutation affecting a core viral process–transcription. It further suggests that multiple infection is likely to modulate the fitness effects of any mutation, such that the topology of fitness landscapes vary with viral population density. Finally, this effect is expected to be most potent under adverse conditions, such as in a new host species.

## Methods

### Cells and cell culture

Madin-Darby canine kidney (MDCK) cells and 293T cells (ATCC, CRL-3216) were maintained in minimal essential medium supplemented with 10% fetal bovine serum and 100 ug/mL normocin. MDCK cells gifted by Peter Palese, Icahn School of Medicine at Mount Sinai were used for experiments. MDCK cells gifted by Daniel Perez, University of Georgia, were used for plaque assays. All cells were maintained at 37˚C and 5% $CO_2$ in a humidified incubator and monitored monthly for mycoplasma contamination.

## Viruses

Influenza A viruses used in these experiments were generated via reverse genetics [27, 28]. Briefly, 293T cells transfected with ambisense plasmids encoding the eight viral gene segments were injected 16–24 h after transfection into the allantoic cavity of 10–11 day-old embryonated chicken eggs (Hyline International) and incubated for 32–36 h at 37˚C. Allantoic fluid was collected and used as the virus stock for experiments. Infectious titers were determined by plaque assay in MDCK cells and by flow cytometry targeting virally encoded epitope tags. Levels of internally deleted defective interfering segments derived from PB2, PB1, PA, and NP segments were confirmed to be minimal for each virus stock using previously-described procedures [29]. Briefly, primers specific to terminal and internal regions of each segment were used in digital droplet PCR. Stocks were deemed low in defective interfering segments if the terminal: internal ratio was below a threshold of 4. All viruses used contained PB2, PB1, HA, NP, NA, M, and NS segments derived from influenza A/mallard/Minnesota/199106/99 (H3N8) virus (MaMN99). MaMN99 PA viruses contained the PA gene segment from MaMN99. MaMN99: GFHK99 PA viruses contained the PA gene segment from influenza A/guinea fowl/Hong Kong/WF10/99 (H9N2) virus. The N-terminus of the HA segment of each virus used was engineered to contain either a 6xHIS or an HA epitope tag connected via a flexible GGGS linker following the signal peptide. This approach was described previously and allows the signal to be present on mature HA proteins without interfering with the folding [30]. Homologous WT and VAR viruses were tagged with opposite epitopes to allow differentiation of infected cells via flow cytometry. GFHK99 Endo PA and MaMN99 viruses were tagged WT as HIS-tag and VAR as HA-tag. All other viruses were tagged WT as HA-tag and VAR as HIS-tag. Homologous VAR viruses included one synonymous mutation in each segment relative to the WT strain, as detailed previously [7].

## Generation of modified PA plasmids

Plasmids encoding chimeric PA gene segments were created using the NEBuilder HiFi DNA Assembly Master Mix (New England Biosciences) according to the manufacturer's instructions using the primers in S1 Table. PCR-amplified fragments encoding the endonuclease region (GFHK99 Endo), linker and arch region (GFHK99 Arch), or C-terminal region (GFHK99 C-term) of GFHK99 PA were combined with a fragment containing the rest of the MaMN99 PA segment in a pDP2002 plasmid, a gift from Daniel Perez [31]. Mutations to the PA 26 codon were introduced using site-directed mutagenesis (QuikChange, Agilent) using primers listed in S2 Table. Two nucleotide changes were introduced to avoid reversion. MaMN99 ΔPA-X and MaMN99:GFHK99 PA ΔPA-X were designed as described previously [16]: site-directed mutagenesis of three bases at the X-ORF (t597c, t600c, and t627a) decrease the likelihood of frameshifting and add a TAG stop codon to truncate the C-terminal region of PA-X. These mutations do not affect the PA coding frame. Sequences of purified plasmid preparations were verified by Sanger sequencing (Genewiz). WT and VAR pairs of each recombinant virus were generated as described above. ΔPA-X segment loss-of-function was verified via host shutoff capacity: 293T cells were transfected with 40 ng of either pCAGGS-MaMN99-PA or pCAGGS-MaMN99-PA-ΔPA-X plasmids and 50 ng of pRL-TK plasmid (Promega). At 24 h post-transfection, the transfected cells were lysed and 20 μl of lysate was transferred to a 96-well plate. 100 μl of *Renilla* luciferase assay reagent (Promega) was added and then *Renilla* luciferase activity was measured on a BioTek Synergy H1 Hybrid Reader. *Renilla* luciferase activity was plotted relative to empty vector transfected cells.

## Infection of cells for quantification of cellular coinfection and viral reassortment

Infections of cultured cells were performed as described previously [7, 32]. Briefly, homologous WT and VAR viruses were mixed in equivalent amounts based on infectious titers as determined by flow cytometry, diluted serially in 1x PBS, and used to inoculate 80% confluent MDCK cells in 6 well dishes. Synchronized single-cycle infection conditions were used: to synchronize viral entry, virus was allowed to attach during a 45 min incubation at 4°C before addition of warm virus medium (1xMEM, 4.3% BSA, 100 IU penicillin/streptomycin) and incubation for 2 h at 37°C. At the end of this 2 h incubation, residual inoculum was inactivated using a 5 min acid wash in PBS-HCl (pH = 3). Cells were then placed in virus medium (pH = 7.1) supplemented with 20 mM $NH_4Cl$ and 50 mM HEPES and incubated at 37°C. This medium prevents endosomal acidification and therefore blocks any further viral entry, imposing single cycle conditions [33, 34]. Released virus and cells were collected at 16 hpi (with 0 hpi defined as the time of warming). Supernatant was stored at -80°C until use in plaque assays.

To verify the effectiveness of our method for limiting an infection to a single round of replication, we set up an experiment in which cells were infected with MaMN99 virus either within medium containing 20 mM $NH_4Cl$ and 50 mM HEPES or with standard medium supplemented with TPCK trypsin (i.e. the medium used to allow 'multi-cycle conditions'). At 24 h post-infection, cell culture supernatant was sampled and infectious titers were determined by plaque assay. Results of this validation experiment showed that, in contrast to the multi-cycle conditions, no viral propagation was detected when single cycle conditions were imposed throughout the inoculation and growth stages of the experiment (S2 Fig).

Coinfections with baloxavir acid (MedChemExpress, CAS No. 1985605-59-1) were completed in the same manner as above, with 5nM BXA added to the virus medium both during the 2 h viral entry period and the 14 h viral replication period.

## Quantification of infection and coinfection

Frequencies of infection and coinfection in cell monolayers co-inoculated with WT and VAR viruses were evaluated based on surface expression of HA- and HIS-tags. Samples were stained for 45 min on ice with Penta HIS Alexa Fluor 647 conjugated antibody (5 ug/ml; Qiagen) and Anti-HA-FITC Clone HA-7 (7 ug/ml; Sigma Aldrich). Cells were then washed and resuspended in PBS-2% FBS in cluster tubes for flow cytometry analysis on a BD-FACSymphony A3 cytometer in the Emory University Flow Cytometry Core. Analysis was performed using FlowJo 10.8.1 software. Non-linear regressions (least squares) and linear regressions of the data were performed in Prism Graphpad.

## Quantification of reassortment

The frequency of reassortant viruses was determined as described previously [35]. Plaque assays were performed in 10 cm-diameter dishes to isolate viral clones and agar plugs were collected with 1 mL serological pipettes into 160 µl PBS. vRNA was extracted using the Quick RNA 96 extraction kit (Zymo) then reverse transcribed using Maxima reverse transcriptase (Thermofisher) per the manufacturer's instructions using the Universal F(A)+6 primer (gcgcgcagcaaaagcagg). cDNA was diluted 1:4 in nuclease-free water and combined with segment-specific primers [7] to differentiate WT and VAR segments by high-resolution melt analysis with Precision Melt Supermix (Bio-Rad) using a CFX384 Touch Real-time PCR detection system (Bio-Rad) and BioRad CFX Manager 3.1 software. Data were analyzed using Precision Melt Analysis 1.3 software (Bio-Rad) to assign a genotype based on the combination of

WT and VAR segments in each isolate. Percent reassortment was calculated as the number of viral isolates with any reassortant genotype divided by the number of isolates screened, multiplied by 100. Results were plotted as a function of percent HA+ cells as determined by flow cytometry. Semi-log curves were fitted to the data in Graphpad.

### Analysis of RNA copy number per infectious unit

Genome copy numbers were determined as follows. RNA was extracted from virus stocks using the QIAamp Viral RNA Mini kit (Qiagen) and reverse-transcribed using Maxima RT (Thermo Scientific) according to the manufacturer's instructions and using the Universal F(A)+6 primer (gcgcgcagcaaaagcagg). Viral cDNA was then quantified by digital droplet PCR using primers targeting the NP segment (MaMN99 NP F: cgacaaagagatcagaagaagga, MaMN99 NP R: tcatcaaatgggtgagacca, GFHK99 NP F: gaaggagagacgggaaatg, GFHK99 NP R: ggctcttgttctctggtatg) and QX200 ddPCR EvaGreen Supermix (BioRad) on a QX200 digital droplet PCR instrument (BioRad). Infectious titers were determined by plaque assay and used to calculate the genome copy:PFU ratio for each sample of virus. Three samples per virus stock were analyzed in this way. The significance of differences between ratios were compared using an unpaired, two-tailed t-test.

### Analysis of the shutoff of cellular gene expression

250, 50, 10, 2, or 0.4 ng of MaMN99 or MaMN99 PA E26K plasmids were ectopically transfected to 293T cells with 50 ng of pRL-TK plasmid using X-tremeGENE 9 (Roche). At 24 h, the transfected cells were lysed and 20 μl of lysate was transferred to a 96-well plate. 100 μl of *Renilla* luciferase assay reagent (Promega) was added and then *Renilla* luciferase activity was measured on a Synergy H1 Hybrid Reader (BioTek).

### Strand-specific quantification of vRNA and mRNA over time

MDCK cells ($1 \times 10^5$ cells per well) were seeded onto 24-well plate and incubated at 37°C for 24 h. The cells were washed three times with PBS, then chilled viruses were inoculated under single cycle infection conditions, low MOI of 0.5 RNA copies/cell and high MOI of 1000 RNA copies/cell. After 1h of absorption, the cells were washed three times with PBS. The cells were collected at 0, 2, 8, and 12 h post-infection, and the RNA was extracted using RNeasy Mini kit (Qiagen). To reverse-transcribe specific RNA species, two different primers targeting vRNA or mRNA of NA segment were used (vRNA primer: ggccgtcatggtggcgaat; mRNA primer: ccagatcgttcgagtcgt) [7]. The quantitative PCR was conducted with Ssofast EvaGreen Supermix (Bio-Rad) and specific primer set targeting vRNA or mRNA of NS using CFX384 Touch Real-time PCR (Bio-Rad) (S3 Table) [36].

### Western blotting

Western blotting was performed using the Thermo scientific miniblot module. SDS-PAGE of reduced samples was run on Bolt Bis-Tris 4–12% premade gels in MOPS. Blotting was performed onto 0.2nm nitrocellulose membranes at 15V for 30 minutes. Blots were blocked with 5% milk in PBS-0.05% Tween 20 for 1h at room temperature, then incubated overnight at 4°C with primary antibodies at 1:2,000: rabbit anti-PA polyclonal (Genetex), mouse anti-β-actin (Sigma, clone AC-74), and mouse anti-influenza NP (Kerafast, clone HT103). Secondary antibodies (1:3,000) goat anti-Rabbit IgG-HRP (Sigma) and goat anti-mouse IgG-HRP (Promega) were incubated with the blots for 1h at room temperature and the blots were developed using

Bio-Rad Clarity Western ECL Substrate. Images were taken on the Bio-Rad ChemiDoc MP imaging system.

## Protein modeling

Models of PA were created using the Phyre2 protein fold recognition server (http://www.sbg. bio.ic.ac.uk/phyre2/) to predict structures based on the published sequence of GFHK99 PA (Genbank accession # MN267497.1) [37]. Visualization was performed with UCSF Chimera (https://www.rbvi.ucsf.edu/chimera/), developed by the Resource for Biocomputing, Visualization, and Informatics at the University of California, San Francisco, with support from NIH P41-GM103311 [38].

## Quantification and statistical analysis

Analysis of these data was performed using the GraphPad Prism statistical software. Replicate sizes are indicated on figure legends where applicable. Linear and non-linear (least-fit) regressions were performed on data collected via flow cytometry and curve similarity assessed via least fit sum-of-squares F test. Semi-log lines were fitted to reassortment data. PA-X host shut-off capacity was analyzed using unpaired Student's t-tests and reported +/- SD for Fig 2A. Data in Fig 5 were analyzed via 2-way ANOVA. Alpha = 0.05.

## Supporting information

**S1 Fig. Example gating strategy for quantification of coinfection frequencies.** The ancestral gates for flow cytometry analysis of coinfection are shown, gating on cells, single cells, and, finally, sorting by epitope tag expression for uninfected, HA-tag+, HIS-tag+, and dual-tag + cells.
(TIF)

**S2 Fig. Validation of inhibition of infection by addition of HEPES buffer and NH$_4$Cl to culture medium.** To confirm that the addition of 20 mM NH$_4$Cl and 50 mM HEPES blocks infection, MDCK cells were inoculated with MaMN99 virus either in standard virus medium supplemented with trypsin or in virus medium lacking trypsin but including 20 mM NH$_4$Cl and 50 mM HEPES. Viral replication was evaluated by titration of released virus sampled at 24 h post-inoculation. N = 3. Significance of differences was evaluated by unpaired, two-sided t-test.
(TIF)

**S1 Table. Primers used for chimeric PA segment Gibson Assembly.**
(DOCX)

**S2 Table. Primers for ΔPA-X site-directed mutagenesis.**
(DOCX)

**S3 Table. Primers used for strand-specific qPCR of MaMN99.**
(DOCX)

**S1 Data. Raw data corresponding to each of the main and supplementary figures and Table 1 are compiled here.**
(XLSX)

## Acknowledgments

We thank Lucas Ferreri for helpful discussion and critical reading of the manuscript.

## Author Contributions

**Conceptualization:** Jessica R. Shartouny, Chung-Young Lee, Anice C. Lowen.

**Data curation:** Jessica R. Shartouny, Chung-Young Lee, Gabrielle K. Delima.

**Formal analysis:** Jessica R. Shartouny, Chung-Young Lee.

**Funding acquisition:** Anice C. Lowen.

**Investigation:** Jessica R. Shartouny, Chung-Young Lee, Gabrielle K. Delima.

**Methodology:** Chung-Young Lee.

**Supervision:** Anice C. Lowen.

**Validation:** Chung-Young Lee.

**Writing – original draft:** Jessica R. Shartouny.

**Writing – review & editing:** Chung-Young Lee, Anice C. Lowen.

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
