## [Decision Letter · Decision Letter 0]

5 Jul 2022

Dear Dr. Lowen,

Thank you very much for submitting your manuscript "Beneficial effects of cellular coinfection resolve inefficiency in influenza A virus transcription" for consideration at PLOS Pathogens. As with all papers reviewed by the journal, your manuscript was reviewed by members of the editorial board and by several independent reviewers. The reviewers appreciated the attention to an important topic. Based on the reviews, we are likely to accept this manuscript for publication, providing that you modify the manuscript according to the review recommendations.

Both reviewers liked the paper but had a few minor concerns that appear worth addressing.

Sincerely,

Christopher B Brooke

Guest Editor

PLOS Pathogens

Carolina Lopez

Section Editor

PLOS Pathogens

Kasturi Haldar

Editor-in-Chief

PLOS Pathogens

orcid.org/0000-0001-5065-158X

Michael Malim

Editor-in-Chief

PLOS Pathogens

orcid.org/0000-0002-7699-2064

Both reviewers liked the paper but had a few minor concerns that appear worth addressing.

Reviewer Comments (if any, and for reference):

Reviewer's Responses to Questions

**Part I - Summary**

Reviewer #1: This manuscript by Shartouny et al. is an interesting follow-up on their prior work indicating that infectious dose could overcome a species barrier for the virus GFHK99, and that the relevant variation to this phenomenon could be found in the PA segment. They find that endonuclease activity, but not specifically the action of PA-X but rather likely the contribution of this activity to mRNA production, is the feature which is compensated by multiple-dose infection. The manuscript is well-written, and other than a few minor suggestions below, the data are easy to follow and well-presented. Overall I would say this is a great contribution to influenza dynamics and the true impacts of deleterious variation in a viral population and the idea that MOI is more than just a number. I also really appreciate the authors providing raw numbers on much of their data, this is a really good thing to do!

My only “real” experimental suggestion is that, given the authors own expertise in working with avian cells, it would be of interest to measure the impact of this variant in avian cells. I admit, my assumption (which could be wrong) from the authors not providing this data is that this variant also is limited in activity in avian cells, which perhaps complicates the host adaptation story. But, if so, I think it is worth pointing out that this mutation could simply be the straw that breaks the camel’s back (ie it is bad in both contexts, but due to other, perhaps smaller effect size, impacts throughout the viral genome, there is a non-linear fitness cliff, below which multiple particles are required to produce robust infection, and above which this effect is marginal, so switching host species reveals pre-existing maladaptive variation simply due to more stringent selection throughout the genome). This is obviously conjecture on my own part, and while I really do think this manuscript would benefit from a simple measurement in this regard, if there are difficulties in procuring that measurement about which I am ignorant I do also feel that the data as presented already present a compelling and interesting story.

Great paper! Looking forward to seeing it in print.

Reviewer #2: Coinfection can modulate viral replication. The authors of this study had previously shown that this was the case for an avian influenza virus, A/guinea fowl/Hong Kong/WF10/99(H9N2) (GFHK99), that was attenuated in mammalian cells and was therefore reliant on the replication boost that came from coinfection. In this follow-on study they seek to understand the mechanism behind this. They map the determinant of attenuation to residue 26 of the viral proteins PA and PA-X, and showed that as the deletion of PA-X did not affect attenuation the primary affect would be in PA. In PA the residue is in the endonuclease domain, and endonuclease ‘up’ mutations at this site also help the virus to overcome the effects of baloxivir marboxil. Both for host-dependent and drug-dependent restriction, more restriction correlates with a greater need for coinfection, which the authors propose is due to achieving a critical level of transcripts in the infected cell.

The experimental work was generally convincing and well-described and the paper clearly written. I found the framing to be a little unexpected – mapping what appears to be a new (if currently rare) host determinant for influenza is interesting and rather underplayed, whereas the framing of the paper as a ‘rare mechanistic insight’ seems a bit overstated. The main finding (which is important on its own terms) seems to be that *any* mechanism that attenuates the virus would lead to an increased reliance on coinfection. This seems important conceptually rather than mechanistically (it could occur through many mechanisms, as indeed it does in this paper and as the authors note in line 307). As a mechanism of PA endonuclease attenuation the paper is really not very detailed, but as a concept the connection between attenuation and coinfection it is important – do the authors think that antiviral treatment might increase the risk of novel viruses emerging through reassortment, for example?

**Part II – Major Issues: Key Experiments Required for Acceptance**

Reviewer #1: 1. Representative (or if possible all) flow cytometry data should be shown to show gating schema.

Reviewer #2: On an experimental note, I am a bit concerned by the way in which MOI was measured – if PA activity is attenuated, HA positivity above a threshold may no longer be a reliable read-out for infection. It would be good to see additional data confirming that for MOIs of GFHK99 and the authors are MaMN99 the authors are in fact comparing like with like.

I also liked the insistence on single cycle infections (line 83) and the method used to achieve this (line 384). However, as the authors note spill-over multi-cycle infections would distort the interpretation of the data, so it would be nice to see data showing that this method for limiting infections to a single cycle is robust.

**Part III – Minor Issues: Editorial and Data Presentation Modifications**

Reviewer #1: 1. As above, a measurement of the impact of this variation in avian cell polymerase assays would be of considerable interest if possible.

2. Looking at the coinfection frequency, it looks non-saturated (ie the number of particles required to produce an HA+ event is less than the number at which coinfection measurements equal 100%). Given that it is non-saturated, my naïve understanding is that, assuming Poisson kinetics, you can place some kind of estimate on the average # of particles required to produce HA+. This might be a really nice number to share with the field. Not necessary, but would definitely be neat to have some kind of estimate.

3. Figure 5 “high” and “low” MOI. The numbers are given in the methods, but why not just write what genome equivalents you are using here in the figure? If not in the figure body, maybe just in the legend?

4. The authors, due to their genome-normalized infection schema, also have a genome/HA+ ratio. I definitely think the reliance on coinfection tells a compelling story, but I also think this ratio would be more immediately graspable in addition to the coinfection measurements. It would be nice if this number was presented to contextualize the coinfection data and show the skew of less infectivity per unit virus (with the coinfection data showing this is due to multiple-dose requirements).

Reviewer #2: General: While it was nice to use a modelled structure for the endonuclease domain of the virus in question, it would also be interesting to discuss this site in the context of the structures of the polymerase trimer during cap-snatching (albeit this would be the homologous site in a different viral strain). Does this give any more mechanistic explanation about what’s going on here – or if not, does it identify a surface to which an as-yet-unidentified host factor might bind? (Or a known factor, such as the ANP32A binding site - do the authors currently have any thoughts on why this site might be host-type dependent?)

Line 10: ‘additional viral templates’ this implies that just adding more vRNA helps, which is an odd mental image – do the authors mean additional RNPs?

Line 113: please clarify how this was done to confirm that this mutation did not affect PA (if indeed this was the case)

Line 136: ‘viruses displayed similar…’ was there any statistical testing? What does similar mean here?

Fig 5G-I: relative to what?

Line 243: ‘lowering the efficiency of viral transcription’ this is true, but there is a connected impact on host shut-off that could also be relevant

Line 248-9: ‘a positive density dependence’ – a bit more discussion would help here. To what extent is there evidence for (more limited) density dependence with WT viruses? What does it mean that the effect of adding more genomes seems greater when the virus is attenuated? Is it that following infection there is a progression towards a point when the cell is saturated, and that if the kinetics of that initial replication are slower there is more scope for additional genomes to make a difference? I appreciate that this might be speculative without further experimental work, but this is the main model that the paper develops and it would be useful to explore its implications in more depth here.

Line 280: The caveats about BXA are probably necessary, but a bit lengthy – I don’t think it’s particularly surprising that viral replication can proceed in the presence of an antiviral drug, and indeed all antiviral drugs tend to have a defined IC50.

Line 343: ‘minimal’ – define (and briefly outline what the procedures were)

Line 348-9: the tag location should be explained more explicitly in a figure or table

Line 384: what is the pH (this is critical)?

Typos, etc

Line 239: ‘basis for these effects are’ -> ‘basis for these effects is’

Line 249: ‘per capita’ – this tends to be used for counting people rather than viruses

PLOS authors have the option to publish the peer review history of their article (what does this mean?). If published, this will include your full peer review and any attached files.

Reviewer #1: No

Reviewer #2: **Yes: **Edward Hutchinson

Figure Files:

Data Requirements:

Reproducibility:

References:

---

## [Editor Report · Decision Letter 1]

8 Sep 2022

Dear Dr. Lowen,

We are pleased to inform you that your manuscript 'Beneficial effects of cellular coinfection resolve inefficiency in influenza A virus transcription' has been provisionally accepted for publication in PLOS Pathogens.

Best regards,

Christopher B Brooke

Guest Editor

PLOS Pathogens

Carolina Lopez

Section Editor

PLOS Pathogens

Kasturi Haldar

Editor-in-Chief

PLOS Pathogens

orcid.org/0000-0001-5065-158X

Michael Malim

Editor-in-Chief

PLOS Pathogens

orcid.org/0000-0002-7699-2064
---

## [Editor Report · Acceptance letter]

14 Sep 2022

Dear Dr. Lowen,

We are delighted to inform you that your manuscript, "Beneficial effects of cellular coinfection resolve inefficiency in influenza A virus transcription," has been formally accepted for publication in PLOS Pathogens.

Best regards,

Kasturi Haldar

Editor-in-Chief

PLOS Pathogens

orcid.org/0000-0001-5065-158X

Michael Malim

Editor-in-Chief

PLOS Pathogens

orcid.org/0000-0002-7699-2064